# Automatic Wheat Ear Counting Using Thermal Imagery

**Jose A. Fernandez-Gallego** [1,2,3], **Ma. Luisa Buchaillot** [1,2], **Nieves Aparicio Gutiérrez** [4],
**María Teresa Nieto-Taladriz** [5], **José Luis Araus** [1,2,*] **and Shawn C. Kefauver** [1,2,*]

1   Plant Physiology Section, Department of Evolutionary Biology, Ecology and Environmental Sciences,
    Faculty of Biology, University of Barcelona, Diagonal 643, 08028 Barcelona, Spain;
    jfernaga46@alumnes.ub.edu (J.A.F.-G.); luisa.buchaillot@gmail.com (M.L.B.)
2   AGROTECNIO (Center for Research in Agrotechnology), Av. Rovira Roure 191, 25198 Lleida, Spain
3   Programa de Ingeniería Electrónica, Facultad de Ingeniería, Universidad de Ibagué, Carrera 22 Calle 67,
    Ibagué 730001, Colombia
4   Instituto Tecnológico Agrario de Castilla y León (ITACyL), Ctra. Burgos Km. 119, 47071 Valladolid, Spain;
    apagutni@itacyl.es
5   Instituto Nacional de Investigación y Tecnología Agraria y Alimentaria (INIA), Ctra. de la Coruña Km. 7.5,
    28040 Madrid, Spain; mtnieto@inia.es
*   Correspondence: jaraus@ub.edu (J.L.A.); sckefauver@ub.edu (S.C.K.);
    Tel.: +34-934021469 (J.L.A.); +34-934021465 (S.C.K.)

**Abstract:** Ear density is one of the most important agronomical yield components in wheat. Ear counting is time-consuming and tedious as it is most often conducted manually in field conditions. Moreover, different sampling techniques are often used resulting in a lack of standard protocol, which may eventually affect inter-comparability of results. Thermal sensors capture crop canopy features with more contrast than RGB sensors for image segmentation and classification tasks. An automatic thermal ear counting system is proposed to count the number of ears using zenithal/nadir thermal images acquired from a moderately high resolution handheld thermal camera. Three experimental sites under different growing conditions in Spain were used on a set of 24 varieties of durum wheat for this study. The automatic pipeline system developed uses contrast enhancement and filter techniques to segment image regions detected as ears. The approach is based on the temperature differential between the ears and the rest of the canopy, given that ears usually have higher temperatures due to their lower transpiration rates. Thermal images were acquired, together with RGB images and in situ (i.e., directly in the plot) visual ear counting from the same plot segment for validation purposes. The relationship between the thermal counting values and the in situ visual counting was fairly weak ($R^2 = 0.40$), which highlights the difficulties in estimating ear density from one single image-perspective. However, the results show that the automatic thermal ear counting system performed quite well in counting the ears that do appear in the thermal images, exhibiting high correlations with the manual image-based counts from both thermal and RGB images in the sub-plot validation ring ($R^2 = 0.75$–$0.84$). Automatic ear counting also exhibited high correlation with the manual counting from thermal images when considering the complete image ($R^2 = 0.80$). The results also show a high correlation between the thermal and the RGB manual counting using the validation ring ($R^2 = 0.83$). Methodological requirements and potential limitations of the technique are discussed.

**Keywords:** thermal images; ear counting; digital image processing; wheat

## 1. Introduction

High throughput plant phenotyping (HTPP) is a quantitative description of the functional and structural properties of the plant [1] for the purpose of crop breeding [2,3]. In the case of cereals,

e.g., wheat, besides grain yield, agronomical yield components are also assessed as part of plant phenotyping pipeline [4]. The accurate quantification of the number of ears per square meter, number of grains per ear and the thousand kernel weight, as the main yield components in wheat, are therefore essential in breeding programs [5]. In the case of ear counting, it is time-consuming and tedious as it is most often conducted manually in field conditions. Moreover, different subsampling techniques and derived protocols for calculation are often used resulting in a lack of standard protocol. As an alternative, several automatic ear counting techniques have been developed in the last years, mainly using as input high resolution RGB (Red/Green/Blue) images. Different image processing techniques have been used such as texture and hybrid color space [6,7], multi-features from color, grayscale and texture data [8]. Decorrelation stretch for color contrast enhancement and Support Vector Machine (SVM) as classification techniques [9,10] and convolutional neural network recognition have been also used [11]. Other approaches use frequency and spatial filter techniques as well as local peaks segmentation [12,13]. Even though the visual spectrum has been widely used for ear counting, there are general limitations to have into account in field conditions, such as solar light conditions (unwanted shadows and bright surfaces), wind conditions (blur ears), ears overlapping and size/shape variation (mostly depending of their more/less horizontal position) and spatial image resolution (camera/canopy distance and sensor size).

Recently, a fusion of multispectral and RGB images have been developed for ear counting estimation [14]. Though not yet applied to wheat ear counting, fruit defect detection using hyperspectral images, through image processing systems, has also been recently developed [15]. Therefore, although visual and multispectral information has been used for ear counting and hyperspectral for fruit defect detection, there is no information in the literature regarding thermal images for ear counting applications or segmentation on such a fine spatial scale, perhaps due to the comparatively low resolution and high cost of thermal cameras [2].

Thermal imagery is related with the transpirative status of the plant [16] that is separate from the visual characteristics that could result in the RGB imagery limitations. Thermal information has been used mainly for monitoring crop water status [17–22] and irrigation management [23,24]. However, previous studies have shown that, regardless of the water conditions during the growing season, there are often significant constitutive differences between leaf and ear temperature on sunny days [25], with ear temperature being higher than leaf temperature. This suggests that thermal imagery may provide a useful approach for ear counting [26]. Temperature distribution across a particular surface have been studied using image processing techniques; for instance, in segmentation applications using the thermal color map, thresholding and morphology operators in research related with orange, apple and almond tree orchards [27–29]. Other similar segmentation approaches have focused on the assessment of plant and leaf temperature separately in vegetables and soybean crops [30,31].

In this study, we propose an automatic wheat ear counting system using thermal images acquired holding the camera by hand above the canopy. We include data captured at three different experimental stations located in northern, central and southern Spain with different environmental conditions. An image processing system was developed to segment the wheat ears taking advantage of the thermal color map. A pipeline structure was designed to filter background and unwanted regions into the image using an adaptive contrast technique and morphological operators. Visual counting directly in the field (i.e., in situ) as well as ear counts derived from RGB images of the same plot segments were also added for the purposes of validating the thermal image and algorithm counting measurements.

## 2. Materials and Methods

### 2.1. Plant Material and Experimental Setup

Two sets of twenty-four post Green Revolution (i.e., semi-dwarf) durum wheat (*Triticum turgidum* L. subsp. *durum* (Desf) Husn.) cultivars (cvs Amilcar, Arcobaleno, Athoris, Avispa, Burgos, Claudio, Core, Don Norman, Don Ricardo, Dorondon, Euroduro, Gallareta, Iberus, Kiko Nick, Mexa, Olivadur,

Paramo, Pedroso, Regallo, Saragolla, Sculptur, Simeto, Solea and Vitron) were grown during two consecutive seasons. In the second year, the Haristide variety was planted instead of Paramo. Field trials were carried at the experimental stations of Colmenar de la Oreja (40°04′ N, 3°31′ W), near Aranjuez (Madrid province) and Coria del Rio (37°14′ N, 6°03′ W), near Sevilla during the 2016/2017 crop seasons and at Zamadueñas (41°42′ N, 4°42′ W), near Valladolid during the 2017/2018 crop season. The first two stations belong to the Instituto Nacional de Investigación y Tecnología Agraria y Alimentaria (INIA), while the third one belongs to the Instituto de Tecnología Agraria de Castilla y León (ITACyL) of Spain. The average annual precipitation and annual temperature is about 425 mm and 13.7 °C, 502 mm and 18.0 °C and 269 mm and 13.2 °C for Aranjuez, Sevilla and Valladolid, respectively. The meteorological data were obtained from the meteorological stations nearest to each experimental station using the SIAR (Sistema de Información Agroclimática para el Regadio) information system [32].

Aranjuez trials were fertilized before sowing with 450 kg ha$^{-1}$ of 8:15:15 (8% N, 15% $P_2O_5$, 15% $K_2O$) fertilizer and in addition 185 kg ha$^{-1}$ of 46% urea was applied before stem elongation. Sevilla trials were fertilized before sowing with 500 kg ha$^{-1}$ of 15:15:15 (15% N, 15% $P_2O_5$, and 15% $K_2O$) fertilizer and 100 kg ha$^{-1}$ of 46% urea was applied before stem elongation. Finally, Valladolid trials were fertilized before sowing with 300 kg ha$^{-1}$ of 8:15:15 (8% N, 15% $P_2O_5$, and 15% $K_2O$) fertilizer; 150 kg ha$^{-1}$ of calcium ammonium nitrate (27% richness in nitrogen) was applied before tillering; and 150 kg ha$^{-1}$ of ammonium sulfate nitrate (26% richness in nitrogen) was applied before heading.

Two experimental conditions (rainfed and supplemental irrigation) were assayed at Aranjuez and Valladolid, while, in the case of Sevilla, only rainfed conditions were assayed. The genotypes were evaluated in 9 m$^2$ in size plots, 6 rows, 0.25 m apart and a planting density of 250 seeds per m$^2$. Randomized blocks were used with three replicates and a total of 72 plots per trial (3 replicates × 24 genotypes). Supplemental irrigation and rainfed trials were planted on 22 December 2016 for Aranjuez and, in the case of Valladolid, supplemental irrigation and rainfed were planted on 13 November 2017 and 23 November 2017, respectively. The rainfed trail at Sevilla was planted on 15 December 2016. Accumulated rainfall and the average temperatures during the crop season for each experimental station were 134 mm and 14.4 °C; 261 mm and 15.7 °C; and 169 mm and 10.2 °C for Aranjuez (2016/2017), Sevilla (2016/2017) and Valladolid (2017/2018), respectively. For field trials under supplemental irrigation, eight irrigations were provided at Aranjuez, with a total of 420 mm of water, and eight irrigations were provided at Valladolid, with a total of 110 mm of water.

*2.2. Thermal Images*

Thermal images were acquired at Aranjuez, Sevilla and Valladolid using the MIDAS 320L infrared camera (DIAS infrared GmbH, Germany) with a −20 °C to 120 °C temperature range, 8–14 µm spectral range in one channel, 320 × 240 radiometric detector and 16-bit format using focal length in manual mode. All files were exported using the default settings for PYROSOFT Professional software (DIAS infrared GmbH, Germany) in BMP (bitmap file) format using 8 bits, and then images were converted to JPG format using 8 bits.

Thermal images from the complete trials (72 plots each) of Aranjuez (supplemental irrigation condition only) and Sevilla (rainfed), together with the first block (24 plots) from Valladolid (rainfed only), were captured for this study. For each plot, one thermal image was taken holding the camera by hand above the canopy and near the center of the plot. Images were acquired after midday in a zenithal/nadir plane at between 0.8 and 1 m distance at each particular growth stage (GS) using Zadoks growth stage [33] (Table 1). Spatial resolution was approximately 0.14 cm/pixel. Images were acquired on 5 May 2017 (10:00–11:00 UTC, GS = 61–65, anthesis), 25 April 2017 (10:00–11:00 UTC, GS = 69, grain filling) and 14 June 2018 (14:30–15:00 UTC, GS = 77, late grain filling) for Aranjuez, Sevilla and Valladolid, respectively. Figure 1 shows an example of thermal images acquired for each experimental station. The actual time of the data acquisition at each location was slightly different to allow for the

adequate contrast between the leaves and ears in the thermal images. Preliminary selection discarded images with acquisition or temperature problems such as blurred images or overcast conditions.

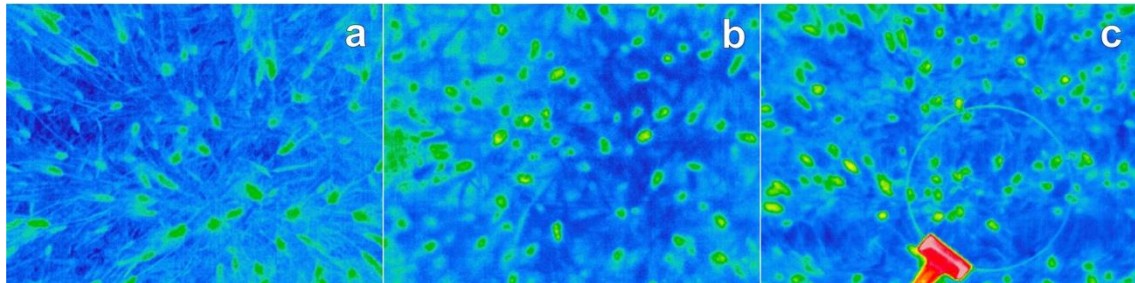

**Figure 1.** Images of plots acquired using the MIDAS 320L thermal camera: (**a**) Aranjuez (anthesis); (**b**) Sevilla (grain filling); and (**c**) Valladolid (late grain filling). The last image includes the ring used for validation purposes.

*2.3. Automatic Thermal Ear Counting System*

This work proposes an automatic image processing system based on the thermal color map using four steps: (1) *low temperature detection*; (2) *contrast limited adaptive histogram equalization* (*CLAHE*); (3) *color threshold*; and (4) *analyze particles* command (Figure 2). The automatic system was developed in ImageJ open source software [34]. As a first step, *low temperature detection* uses the CIE L*a*b* color space [35] to avoid the blue color values; the negative b* values were filtered using the *color threshold* macro [34]. *CLAHE* method [36] was used to enhance the local contrast in small regions in the image. As a next step, *color threshold* macro was used to select the high temperature via the Hue/Saturation/Value (HSV) color space [37], represented in colors between red and green, which correspond to hue values from 2 to 120, and therefore closely related with the presence of ears. Finally, *analyze particles* function [34] was used to count and filter the regions detected as ears.

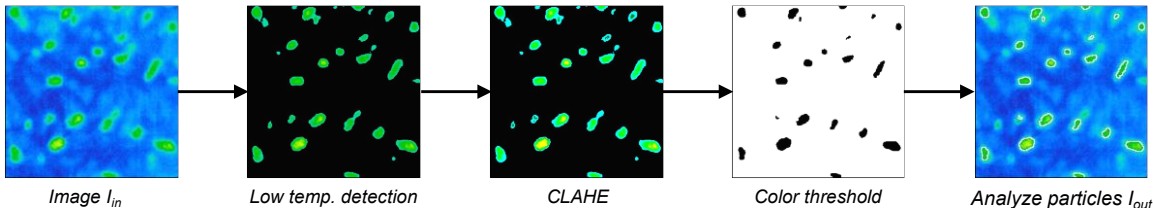

| Image $I_{in}$ | Low temp. detection | CLAHE | Color threshold | Analyze particles $I_{out}$ |

**Figure 2.** Automatic thermal ear counting system: (1) *low temperature detection*; (2) *contrast limited adaptive histogram equalization* (*CLAHE*); (3) *color threshold*; and (4) *analyze particles* command, boundaries regions detected as an ear were underlined in white color.

Color thermal maps were used for the ear detection system, and the CIE L*a*b* color space was selected with the aim of detecting the lower temperatures in the image. This color space uses a Cartesian system of coordinates, where the positive b* axis represents the amount of yellow and the negative b* axis represents the amount of blue [35]; in that way, we filtered the negative b* axis to avoid leaves, which are related with lower temperature. The a* axis was not filtered for this step. The *CLAHE* algorithm was used to enhance the local contrast in edges and regions into the image and contribute to isolate overlapping or neighboring ears. The HSV color space uses the hue values from 0° to 360° to represent colors from red to magenta, while saturation and value (or brightness) have numbers from 0 to 100 [37]. This color space was used to segment high temperature represented in colors between green and red. Finally, *analyze particles* command was used to count and filter the regions detected as ears by the automatic algorithm.

*2.4. Algorithm Validation*

Manual In Situ Counting and RGB Images

For validation purposes, a physical ring was placed on the top of the canopy for counting the number of ears in the exact ring area by visual inspection in the field. The ring has a radius of 0.1225 m. The ring was attached by an extension arm to the monopod used to acquire the RGB images. Thermal and RGB images were acquired at the same time as the visual (in situ) ear counting (inside the ring) was assessed in the first block (i.e., 24 plots) of the rainfed trial at Valladolid. Visual counting was always performed by the same person at the same position where the images were acquired. Approximately 15 s were spent for each counting using a clicker to keep track of the exact number and making sure to inspect the area inside the ring to accurately include all ears present by moving plants and changing perspective angles at each location. Additionally, RGB images from the same plot segments were acquired (at the same time than the thermal images) in a zenithal/nadir plane with a Sony QX1-ILCE camera (Sony Corporation, Japan), 20.1-megapixel resolution, with 23.2 × 15.4 mm sensor size, using 16 mm focal lens and resolution of 5456 × 3632 pixels. The images were taken using a monopod at 1 m above the canopy. The resulting RGB image spatial resolution was approximately 0.03 cm/pixel.

The presence of an ear inside the ring area assigned through the thermal images was checked by the RGB image (Figure 3) together with the in situ visual counting. In that way, it was assured that the temperature changes were due to the presence of an ear instead of soil, leaves or unwanted objects.

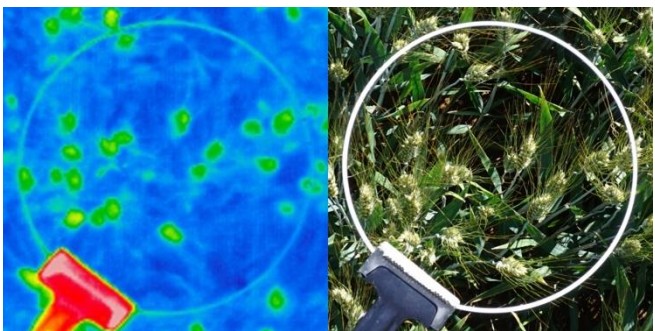

**Figure 3.** Thermal and RGB images were acquired for Valladolid at late grain filling. A ring was used as a reference area for validation purposes. The number of ears inside the ring area were counted using the thermal and the RGB images and, additionally, the number of ears was counted by visual inspection in the field. The black extension-arm that supported the ring showed higher temperature than ears and canopy (in red color), enabling it to be automatically extracted by morphology operators in the image processing system.

Two validation steps were developed using manual image-based counting. On the one hand, the ears inside the ring area (including the ring edge) in the thermal and RGB images were manually marked, and the visual ear counting data from the field and the algorithm results were also included. The results are referred to as *Ring-Manual-In-situ-Counting* (*Ring-MIC*), *Ring-Manual-Thermal-Counting* (*Ring-MTC*), *Ring-Manual-RGB-Counting* (*Ring-MRC*) and *Ring-Algorithm-Thermal-Counting* (*Ring-ATC*). A set of 24 images and full counting datasets were used for each variable of the ring related measurements, thermal and RGB images. On the other hand, selected additional complete (full-sized) thermal images (without cropping to the size of the reference ring) were also manually marked (Figure 1). The result is referred to as the *Complete-Manual-Thermal-Counting* (*Complete-MTC*). Finally, the number of ears automatically detected by the algorithm is referred to as the *Complete-Algorithm-Thermal-Counting* (*Complete-ATC*). The ears manually marked in the images were counted using a simple algorithm developed for counting the number of colored marks present in the image. The markers were placed using the *Pencil tool* [34] with the same color value and circular

shape and size. In this way, the simple algorithm for the marker counting could be limited to search for precisely the same color and shape marks to then segment and count them.

### 2.5. Statistical Analysis

Data analysis was performed using the open source software, RStudio 1.1.423 (R Foundation for Statistical Computing, Vienna, Austria). Lineal regressions were used to analyze the relationship between manual image-base counting and automatic thermal ear counting. The data were plotted using SigmaPlot version 12 (Systat Software, Inc., San Jose, CA, USA).

## 3. Results

### 3.1. Linear Regression between Thermal, RGB, In Situ and Algorithm Counting

Linear regression of *Ring-MIC*, *Ring-MRC* and *Ring-MTC* against *Ring-ATC* was calculated for the 24 rainfed plots from Valladolid at late grain filling growth stage (Figure 4). The relationships between *Ring-ATC* against *Ring-MIC* ($R^2 = 0.40$), *Ring-MRC* ($R^2 = 0.84$) and *Ring-MTC* ($R^2 = 0.75$) were positive and statistically significant ($p$-value < 0.001). Therefore, the weakest correlation was recorded against the visual counting in the field, which a priori represents the actual number of ears present. We also calculated the relationship between the thermal and the RGB manual counting using the ring, where a positive correlation with statistical significance was obtained ($R^2 = 0.83$, $p$-value < 0.001). In addition, the relationship of *Ring-MRC* against *Ring-MIC* was positive and statistically significant ($R^2 = 0.37$, $p$-value < 0.001), and similar in strength to the correlation between *Ring-ATC* and *Ring-MIC*.

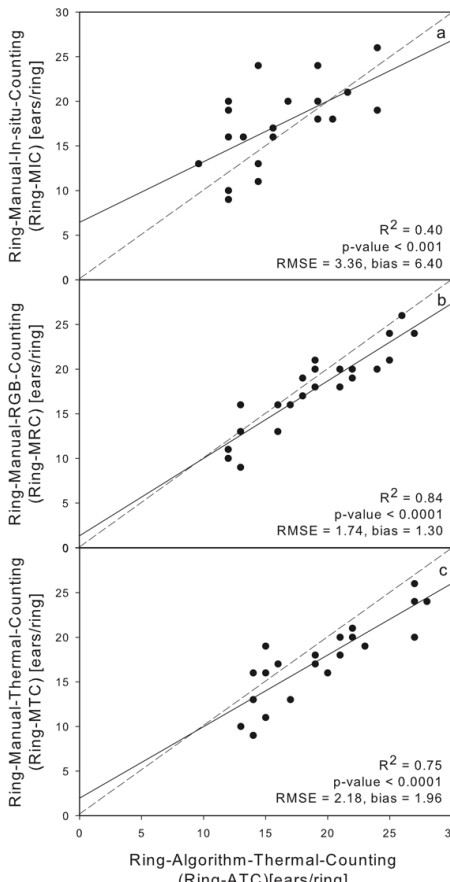

**Figure 4.** Linear regression of the relationships using the ring area for: (**a**) *Ring-MIC* ($R^2 = 0.40$); (**b**) *Ring-MRC* ($R^2 = 0.84$); and (**c**) *Ring-MTC* ($R^2 = 0.75$) vs. *Ring-ATC* using images from Valladolid rainfed trial at late grain filling. The dotted lines indicate the 1:1 slope.

On the other hand, the relationship using the preselection of complete (full-sized) thermal images from Aranjuez, Sevilla and Valladolid against the algorithms counting (*Complete-ATC* vs. *Complete-MTC*) were also positive, statistically significant ($R^2$ = 0.80, *p*-value < 0.001) and close to 1:1 slope relationship (Figure 5).

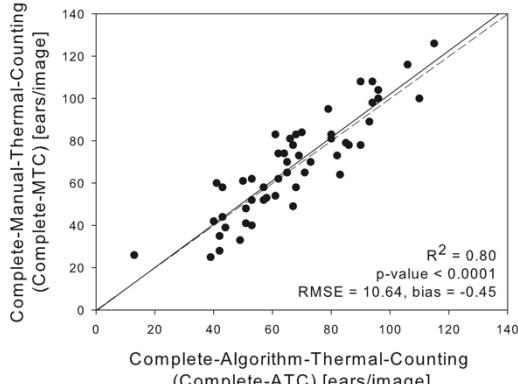

**Figure 5.** Linear regression for *Complete-MTC* vs. *Complete-ATC* ($R^2$ = 0.80) using the full-sized thermal images from Aranjuez, Sevilla and Valladolid at anthesis, grain filling and late grain filling, respectively. The dotted line indicates the 1:1 slope.

### 3.2. Understanding Acquisition and Algorithm Errors

Figure 6 shows three temperature image scenarios related with the acquisition protocol, wheat crop temperature and optimal algorithm considerations. The image in Figure 6a was acquired at around 10:30 UTC in Aranjuez (supplementary irrigation) when, at this time, the ears exhibited higher temperature than the canopy leaves due to direct sunlight conditions for several hours. In the case of low thermal image contrast (Figure 6b), there were no temperature differences observed between the ears and the rest of the canopy due image acquisition at 10:30 UTC in Sevilla, when overcast conditions inhibited any direct sunlight to increase ear temperatures. Thus, the ears could not be detected separately by the temperature sensor, resulting in some leaves being detected as an ear by the algorithm. On the other hand, Figure 6c shows an image acquired in Aranjuez at the same optimal daytime as Figure 6a, although the acquisitions distance used was less than 0.8 m by mistake, so that the ears visually appear blurred in the image; the algorithm could not isolate property the ear regions. The images in Figure 6b,c may be considered as acquisition errors due to improper sky conditions and camera user error, respectively. Providing adequate sky conditions and correct camera settings, the algorithm errors related with ear identification are relatively minor (Figure 6a) and basically due to the inability of the algorithm to detect or separate very closely or overlapping ears in these circumstances (see red color semi-circle in Figure 6a; similarly, two ears were not identified, as shown by yellow circles in dots in Figure 6a) by the algorithm due to the lack of contrast.

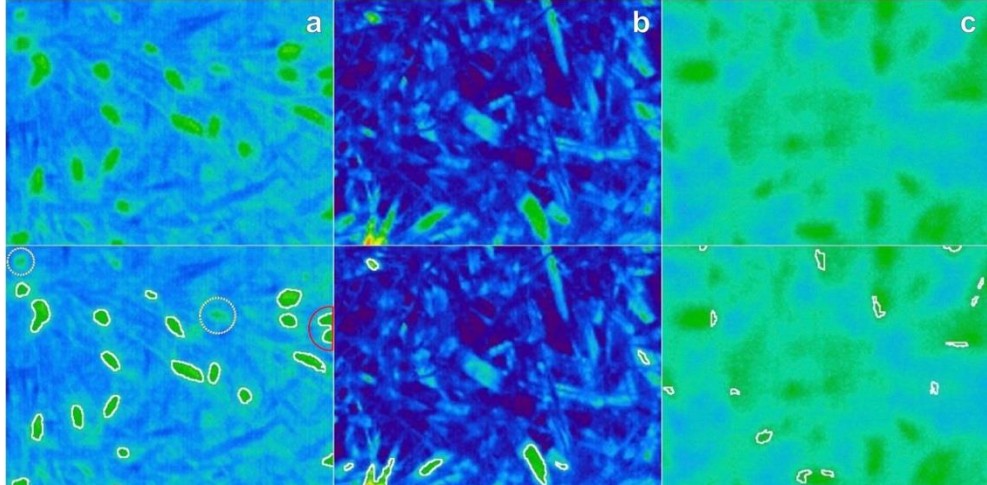

**Figure 6.** Thermal images: (**a**) optimal temperature: higher ear temperature than canopy temperature; (**b**) low thermal image contrast: no temperature differences between canopy and ears; and (**c**) out-of-focus image: ears and canopy at high temperature, and the image was acquired at less than 0.8 m distance between the camera and the canopy. The boundary regions underlined in white color represent the ears automatically detected by the algorithm.

## 4. Discussion

Ear density can be used as a target breeding trait in cereal phenotyping. To date, the few studies dealing with automatic ear counting in the field have mostly been performed using RGB images [6,8–13]. Besides the intrinsic low cost of this approach due to the easy operation and affordability of digital cameras, the high resolution of the natural color digital images is a major factor to consider as both a cost and a benefit. The use of RGB images may have limitations under certain field conditions, including the quality of the sky and light conditions, which can be overcome with sufficiently high spatial resolution, but which requires powerful computing capacities and makes its implementation more complex or less high throughput than expected. Other remote sensing approaches include the use of multispectral images [14], but the segmentation accuracy decreases as the canopy area observed within a single image increases, potentially due to the lower spatial resolution of these images and the reflectance angle dependence of multispectral data. Even LIDAR may be used [38] but its price and processing requirements may still be considered prohibitive and its size and weight makes it too cumbersome to be handheld or pole mounted for quick ground evaluation in field conditions. As an alternative, thermal images may be used. While thermal imagery may provide slightly lower spatial resolution compared to multispectral images, the possibilities for obtaining images with a much greater contrast between ears and leaves is much higher with thermal imaging. The increase in contrast provided by thermal imaging stems from large differences in ear and leaf transpiration rates, which directly affect cooling capacity and temperature. To ensure differences in temperature between the ears and the rest of the canopy, it is still recommended the images be acquired within a few hours of solar noon to reduce shadowing and sun angle effects. In fact, this recommendation may be extended for any passive remote sensing imaging technique.

Moreover, thermal cameras use radiation far from the visible and near infrared spectral regions and thus the factors that contribute to some of the limitations of RGB digital images such as brightness of other factors affecting light conditions [2,16,26] are removed. For that reason, thermal images have proven to be easier to process than RGB images, in part due to their lower resolution, without the lack of contrast and technical limitations of multisensory array multispectral imagers. In fact, the ears, and regardless of the water conditions, are usually several degrees warmer than the leaves [25], due to their constitutive lower stomatal conductance and thus transpiration rates compared with the leaves [39]. For this study, we measured the leaf and the ear temperature, with differences

within images ranging 1.9–5.0 °C, 2.1–3.4 °C and 2.0–5.0 °C for Aranjuez, Sevilla and Valladolid, respectively. The mean temperate differential between ears and leaves across all treatments needed for the algorithm to segment property the ears was around 2.0 °C. Although, for this application, we used the thermal color map to focus on the contrast present due to relative temperature differences, it is also possible to work on the full radiometric kelvin information to get, for instance, the mean, range or specific ear(s) temperature from the thermal images using the same segmentation algorithm. It could represent additional useful information for phenotyping tasks as ear temperatures have been reported in some cases to be better correlated with grain yield than spectral vegetation indices and also provide comparable correlations as gene expression performance in predicting grain yield [25].

For additional thermal image algorithm validation purposes, visual in situ counting was developed using a ring to delimit a specific area over the crop while in the field and thus facilitate the manual counting. Although the ring has a small area, compared with the complete plot size, we obtained a relatively low $R^2$ relationship against thermal image-based counting ($R^2$ = 0.40, Figure 4). This is most likely associated with the limited single image-perspective of the one zenithal/nadir thermal image or RGB image captured in the field. Some portion of the error could additionally be associated with human visual inspection errors in the field and potentially the subjectivity of the observer, as are often assumed to be major sources of error in manual ear counting in actual breeding programs; however, for this study, the researchers attempted to minimize the human error associated with the *Ring-MIC* ear counts to provide quality validation data. In the manual in situ counting in the field, it was necessary to both view the canopy from different angles as well as physically move plants to acquire accurate field validation data, representing a major difference between the in situ counting and the single image-perspective remote sensing approach of the automatic thermal image ear counting technique presented here. In previous studies on ear recognition, no information regarding the correlation between in situ visual ear counting and automatic ear counting was provided [6–14], but it is nonetheless an important point to consider as the entire image acquisition and processing pipeline represents a sum of errors. Of course, the approach for visual counting assayed was in fact much faster than the traditional ear counting procedures, which implies for example counting the total number of ears in one-meter linear row length. However, this approach is quite tedious (and of course takes much longer than the 15 s per plot as in our study). Nonetheless, we obtained good results using thermal imagery for ear counting with positive and strong relationships between the automatic thermal ear counting system and the manual image-based ear counting ($R^2$ = 0.84 for *Ring-MRC*, Figure 4; $R^2$ = 0.75 for *Ring-MTC*, Figure 4 and $R^2$ = 0.80 for *Complete-MTC*, Figure 5). Furthermore, in all comparisons, the slope of the correlation was quite close to a 1:1 ratio, indicating very little bias toward over- or under-counting within the range of ear density in this study. Thus, the additional validation results provide support for the capacity of the automatic thermal image counting algorithm to count the ears that are present in the image with high precision and low bias. However, other potential sources of error in the thermal image counting pipeline should be considered in more detail.

Although we also detected limitations specific to the use of thermal imagery for ear counting, such as the observed crop temperature issues (Figure 6), there are also errors related to the general use of remote sensing imaging for ear counting, potentially applicable to any other single image or "snapshot" approaches regardless of the range of non-penetrating electromagnetic radiation employed. This is the case for instance of overlapping and hidden ears and might explain the rather low correlation between the *Ring-MRC* (single human eye perspective) and the *Ring-MIC* ($R^2$ = 0.37). The use of additional oblique/off-nadir thermal imaging may provide improved canopy penetration, as for instance 3D surface models that suggest performance improvements when off-nadir images are incorporated [40,41], but may also come with other complications (consistency in oblique off-nadir angle, determination of optimal angle, and more complex 3D processing algorithms) or yet unknown errors.

**Table 1.** Comparative thermal and RGB data information in field conditions for ear counting applications.

|  | Thermal | RGB |
|---|---|---|
| Temperature of the Ears | Several degrees warmer than leaves [25]. | Irrelevant. |
| Stage growth | From heading to near crop maturity [42]. | From heading to near crop maturity [12]. |
| Accuracy hour of the day and sky conditions | Clear sky conditions. After midday until 18:00, depending on plant water stress conditions. | Depends of the hour of the day, 8:00 to 18:30 [10], 8:00 to 17:00 [14], 9:00 to 16:00 [9], 12:00 to 16:00 and sky (preferably diffuse light) conditions [12]. |
| Position of the camera | Zenithal/nadir. | Zenithal/nadir [6,10–13]; 45° above the horizontal [14]. |
| Distance of the camera from crop | 0.8–1 m. | 0.85 m [6], 2.5 m [10], 2.9 m [11], 3.5 m [14], 0.8–1 m [12]. |
| Spatial resolution from ground acquired images | Approximately 0.14 cm/pixel, depending on camera and distance from crop. | Ranging 0.01–0.25 cm/pixel [6,10–13]; depending on camera and distance from crop. |
| Possible algorithm errors | -The algorithm presents errors when the air temperature is too low or high or the sky is too cloudy, or the conditions very windy, which may prevent differences between the canopy and the ear temperature appears. -The camera could be out-of-focus, potentially due to a very short image acquisition distance between the camera and the canopy. -In sparse canopies, soil temperature may affect the background. -Dry or senescent leaf canopy may affect the background. | -False positives where pixels are labeled as ears correspond to leaves and result in irregularities in the ear counting. -False negatives result in ears that are not detected by the algorithm because the contrast between the ear and soil is not great enough and the segmentation algorithm discarded that region. -The algorithm labeled the area as an ear, where the pixels are soil and noise being a result of background brightness caused by a foreign object [12]. |

Thermal and RGB image data in field conditions are discussed throughout this work and each of the technology, acquisition and image processing steps show some limitations (summarized in Table 1). RGB sensors provide high frequency information (very high spatial resolution) that contributes to improvements in perceiving the existence of an ear separate from leaves, soil and other unwanted objects; even though similar texture characteristics can be found in the awns or in parts of the leaves for instance due to the high RGB resolution [12], these similarities between awns and ears actually increases the challenges for automatic RGB ear counting systems. On the other hand, thermal images filter high frequency details intrinsically due to the different technology that it uses to detect much longer wavelength radiation emissions and its low-resolution characteristics [16]; this helps in the automatic ear counting system implementation using thermal images. However, we could detect some similar RGB and thermal image errors, such as overlapping and non-identified ears, yet they may still provide some complimentary benefits together, such as more flexible image acquisition conditions or improved image feature extraction and opportunities for validation. Therefore, thermal and RGB fusion may in combination provide the best features of each technology in a way that could be acquired by new mobile phones that incorporate thermal sensors [2]. Even more advanced systems that include hyperspectral cameras [15] may also be considered in the future for ear counting purposes.

On the other hand, researcher visual interpretation of the RGB images was crucial in correctly locating the presence of ears in thermal images for the development of the thermal image ear counting algorithm. Thereby, the acquisition of thermal and RGB images at the same time may contribute further to the understanding and interpretation of the information in the thermal images, contributing to the development of a more robust algorithm for ear counting with thermal image color maps. In fact, the ears are usually several degrees warmer than leaves for only some parts of the day under the right conditions; thus, for ear counting purposes, it is necessary to select the optimal time of day for acquiring thermal images. The thermal images, when taken at the right moment, can provide, from an image processing system perspective, clearer information of the different components of the canopy;

however, in some cases, high temperature information could be associated with soil or unwanted objects that RGB images can help to avoid. Therefore, for future work, it may be the case that perhaps thermal and RGB fusion could be the next step for ear counting applications.

## 5. Conclusions

In our study, a thermal camera was used to develop an image processing system for automatic ear counting in field conditions. In favor of the thermal counting approach, ear density values estimated through thermal imaging can be processed much more rapidly as the size of the images is much smaller compared to high resolution RGB images used in other previous studies, while the increase in contrast allows for equally accurate assessments when the thermal images are captured under specific conditions. There should be a difference of at least 2 °C between the ear and leaf temperatures for this thermal ear counting algorithm to work. Although the correlation with manual in situ ear counts (*Ring-MIC*) was not very high, the algorithm did demonstrate high correlations with various manual image-based ear counts (*Ring-MRC*, *Ring-MTC*, *Complete-MTC*). In future applications, thermal imagery may be acquired from multiple perspectives (including off-nadir and oblique), or even thermal video data, for improved ear detection in comparison with in situ counts. However, further studies could use the same thermal image segmentation algorithm developed here for ear detection (Figure 2) to extract the temperature of ears and leaves separately for other phenotyping applications related to plant water stress effects or grain yield prediction. Thermal and RGB fusion, along with 3D imaging, could be the next steps for cereal ear counting in field conditions to take maximal advantage of the strengths of each imaging technology.

**Author Contributions:** Conceptualization, J.A.F.-G., M.L.B., J.L.A. and S.C.K.; methodology, J.A.F.-G., M.L.B., N.A.G., M.T.N.-T., J.L.A. and S.C.K.; software, J.A.F.-G.; validation, J.A.F.-G. and M.L.B.; formal analysis, J.A.F.-G., M.L.B., J.L.A. and S.C.K.; investigation, J.A.F.-G., M.L.B., N.A.G., M.T.N.-T., J.L.A. and S.C.K.; resources, N.A.G., M.T.N.-T., J.L.A. and S.C.K.; data curation, N.A.G. and M.T.N.-T.; writing—original draft preparation, J.A.F.-G., M.L.B., J.L.A. and S.C.K.; writing—review and editing, J.A.F.-G., J.L.A. and S.C.K.; visualization, J.A.F.-G. and M.L.B; supervision, J.L.A. and S.C.K.; project administration, J.L.A.; funding acquisition, J.L.A.

**Funding:** This work was supported by MINECO, Spain (project number AGL2016-76527-R) as the primary funding support for the research project; the project "Formación de Talento Humano de Alto Nivel" (project number BPIN 2013000100103) approved by the "Fondo de Ciencia, Tecnología e Innovación", from the "Sistema General de Regalías"; and "Gobernación del Tolima—Universidad del Tolima, Colombia" as the sole funding source of the first author J.A.F.-G. Contribution of J.L.A. was supported in part by ICREA Academia, Generalitat de Catalunya, Spain.

**Acknowledgments:** The authors of this research would like to thank the field management staff at the experimental stations of Colmenar de Oreja (Aranjuez) and Coria del Rio (Sevilla) of the Instituto Nacional de Investigación y Tecnología Agraria y Alimentaria (INIA) and Zamadueñas (Valladolid) of the Instituto de Tecnología Agraria de Castilla y León (ITACyL) for their continued support of our research.

**Conflicts of Interest:** The authors declare that they have no competing interests.

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
