# Peer review of "Automatic Wheat Ear Counting Using Thermal Imagery"

_remotesensing, doi:10.3390/rs11070751_

Round 1

Reviewer 1 Report

Line 141-142: it would be useful to give an idea of what acquisition or temperature 'problems' were encountered, in particular their frequency because it would provide useful information about the applicability in breeding plot scenarios.

Figure 3: How are ears that fall right on the edge of the validation ring counted (or not)? In this Figure there are a couple of these cases. It would be useful to clarify this aspect because it may have an undisired impact on the comparison. Line 196 - 197 seems to suggest that ears overlapping with the edge of the circle are not counted. However, it would be good to spell this out clearly.

Line 205-206 'simple algorithm'. Can a reference be given or a short explanation?

Figure 4, 5. The regression lines have an intercept on the y-axis (also in Figure 5?). Is there a rational for not having the regression line constrained to zero on the y-axis? Moreover, the legend inside the graphs (p-values) should be changed from Spanish to English.

Line 275-278. Consider rephrasing, sentence is unclear.

Line 299-303. The argumentation leads to the point that, in fact, that the visual counting in the field used in this study is not a very reliable way to assess 'ground truth'. Specific comments on how to further validate the methodology would be helpful.

Author Response

Response to Reviewer 1:

Comments to the Author

Line 141-142: it would be useful to give an idea of what acquisition or temperature 'problems' were encountered, in particular their frequency because it would provide useful information about the applicability in breeding plot scenarios. 

We have added and additional sentence to provide more information related with temperature problems.

Figure 3: How are ears that fall right on the edge of the validation ring counted (or not)? In this Figure there are a couple of these cases. It would be useful to clarify this aspect because it may have an undesired impact on the comparison. Line 196 - 197 seems to suggest that ears overlapping with the edge of the circle are not counted. However, it would be good to spell this out clearly. 

To clarify this important point for validation purposes, we have added a sentence at the same manuscript line to establish the ring edge as part of the ring area.

Line 205-206 'simple algorithm'. Can a reference be given or a short explanation?

We have added and additional sentence to explain the ImageJ tool used for the manual image-based counting and the automatic algorithm specifications as follows:

The ears manually marked in the images were counted using a simple algorithm developed for counting the number of coloured marks present in the image. The markers were placed using the Pencil tool [34] with the same colour value and circular shape and size. In this way, the simple algorithm for the marker counting could be limited to search for precisely the same colour and shape marks to then segment and count them.

Figure 4, 5. The regression lines have an intercept on the y-axis (also in Figure 5?). Is there a rational for not having the regression line constrained to zero on the y-axis? Moreover, the legend inside the graphs (p-values) should be changed from Spanish to English.

We have modified the Figures 4 and 5 starting from (0.0) and changed the legend to ‘p-value’. We have also added the 1:1 line in each regression figure and, in the figure legend, we have added the RMSE and intercept/bias values.

Line 275-278. Consider rephrasing, sentence is unclear.

We have rephrased the sentence as follows for simplicity and clarity:

The photosynthetic pigments of ears and leaves as assessed in the field using a multispectral camera is much more similar than the differences in temperature between both organs as detected by a thermal camera, which is related to their similar photosynthetic activity but very different rates of transpiration. In order to ensure differences in temperature between the ears and the rest of the canopy it is still recommended the images be acquired within a few hours of solar noon in order to reduce shadowing and sun angle effects. In fact, this recommendation may be extended for any passive remote sensing imaging technique.

Line 299-303. The argumentation leads to the point that, in fact, that the visual counting in the field used in this study is not a very reliable way to assess 'ground truth'. Specific comments on how to further validate the methodology would be helpful.

The Referee is right and in fact this was one of the points stressed by the study. Nevertheless, and following the suggestion of the Reviewer the second sentence (of these lines) was modified completely in order to expand the visual in-situ counting protocol developed in the field. We have also added new reasoning regarding human vision inspection and RGB-image perspective and have incorporated it into the Discussion and Conclusions accordingly.

In the methods section:

For validation purposes, a physical ring was placed on the top of the canopy for counting the number of ears in the exact ring area by visual inspection in the field. The ring has a radius of 0.1225 m. The ring was attached by an extension arm to the monopod used to acquire the RGB images. Thermal and RGB images were acquired at the same time than visual (in-situ) ear counting (inside the ring) were assessed in the first block (i.e. 24 plots) of the rainfed trial at Valladolid. Visual counting was performed always by the same person at the same position where the images were acquired. Approximately 15 seconds were spent for each counting using a clicker to keep track of the exact number and making sure to inspect the area inside the ring in order to accurately include all ears present by moving plants and changing perspective angles at each location.”

In the Discussion:

This is most likely associated with the limited single image-perspective of the one zenithal/nadir thermal image or RGB image captured in the field. Some portion of the error could additionally be associated with human visual inspection errors in the field and potentially the subjectivity of the observer, as are often assumed to be major sources of error in manual ear counting in actual breeding programs; however, for this study the researchers have attempted to minimize the human error associated with the Ring-MIC ear counts in order to provide quality validation data. In the manual in-situ counting in the field, it was necessary to both view the canopy from different angles as well as physically move plants in order to acquire accurate field validation data, representing a major difference between the single image-perspective remote sensing approach of the automatic thermal image ear counting technique presented here. In previous studies on ear recognition, no information regarding the correlation between in-situ visual ear counting and automatic ear counting was provided [6–14], but it is nonetheless an important point to consider as the entire image acquisition and processing pipeline represents a sum of errors. Of course, the approach for visual counting assayed was in fact much faster than the traditional ear counting procedures, which implies for example counting the total number of ears in one-meter linear row length. However, this approach is quite tedious (and of course takes much longer than the 15 s per plot as in our study).

In the Conclusions:

Though the correlation with manual in-situ ear counts (Ring-MIC) was not very high, the algorithm did demonstrate high correlations with various manual image-based ear counts (Ring-MRC, Ring-MTC, Complete-MTC). In future applications, thermal imagery may be acquired from multiple perspectives (including off-nadir and oblique), or even thermal video data, for improved ear detection in comparison with in-situ counts.

It our hope that the Reviewer finds that these changes with regards to each specific point have resulted in an acceptable manuscript for publication in this special issue of Remote Sensing.

Reviewer 2 Report

See attached file.

Author Response

Response to Reviewer 2:

Comments to the Author

The article deals a very interesting subject, the possibility of counting wheat ears using TIR cameras in an automatic way, opening the possibility of substitute the usual form of counting, manual counting in situ, by the new proposed method/algorithm. However, it is clear from the result when comparing both methods (R2=0.40) that, at least in this case, the proposed method has not been useful. I think that a bad result is also a result and it must be considered to be published, but the authors must be honest and present the reality, i.e., that the result is bad and not to trying to hide this. So, trying to hide the obvious, the article becomes increasingly confusing and, at the end, one does already not know which was the objective of the article and even which were the reference values, which supposedly should be the manual counting in the field (Ring-MIC). A lot of work has been made in the field and with the images processing, but the article must present the results correctly and clarify other aspects, if not, I will reject it definitely. Besides, I do not agree with some explications to the results given by the authors.

We agree with the Reviewer that there is one weak result in the primary validation of the study, namely the utility of the technique and algorithm pipeline for counting ears in the field. Consequently, we have changed much of the text to make it more clear that we are not trying to “hide” this negative result in any way. There are several other positive results afterwards that demonstrate that the algorithm developed and presented here does work quite well in analyzing the images taken in the field and we have included those results as well. As a result, we have changed much of the results comparison with a more thorough perspective and in-depth critique with regards to the functionality of the entire data acquisition and processing presented here. Moreover, we have made changes in accordance within the introduction, methods, results and most importantly, the discussion and conclusions. We have added some ideas for future improvements based on this detailed assessment of errors as well as the potential for expanded uses of the image processing algorithm as it is. As many of these are answered separately, we provide the details in continuation below.

Main problems to correct/clarify:

Abstract: From the last paragraph (lines36-38) is deduced that the automatic system proposed is an advantage respect to the in-situ visual counting, and it is not true, because the in-situ data are always considered as the reference values and one validate its results with them. If the references values are not good, then the authors have a problem, because then they do not have reference values. They should clarify this. Besides, if they did not consider that the manual counting with only 15 seconds was good, then they could have made a good counting spending more time, because the exact and real amount of ears in each ring is the important value here. After, they could also have validated the protocol of 15 sec. with the exact measurement.

We have redirected completely the abstract structure locating the manual in-situ counting (Ring-MIC) as a main reference point. We have also added correlation results regarding automatic and manual image-based counting for thermal and RGB images as well as the correlation between thermal and RGB manual counting using the ring. The entire second part of the abstract has been rewritten as follows in order to more directly note all of the relevant results:

“The relationship between the thermal counting values and the in-situ visual counting were weak (R2 = 0.40), which highlights the difficulties in estimating ear density from one single image-perspective. However, the results show that the automatic thermal ear counting system performs quite well in counting the ears that do appear in the thermal images, exhibiting high correlations with the manual image-based counts from both thermal and RGB images in the sub-plot validation ring (R2 = 0.75-0.84) and with the manual counting from thermal images when considering the complete image (R2 = 0.80). The results also show a high correlation between the thermal and the RGB manual counting using the validation ring (R2 = 0.83). Methodological requirements and potential limitations of the technique are discussed.”

The manual in-situ counting protocol was described in more detail in Material and Methods (subsection 2.4.1. Manual in-situ counting and RGB images) to clarify the process carried out in the field.

So, are the Ring-MIC values the reference values or not?

Yes, the Ring-MIC values are the reference values for the results of the entire image acquisition and processing pipeline together and the primary goal of the study. That said, the Ring-MIC values (as validation) is the final validation for the whole procedure, but in the assessments of the algorithm function itself we consider the references to the image-based manual counting to be of greater importance for the consideration of the assessment of algorithm alone. As such, we are left with the remaining error belonging to some part of the whole pipeline; in this case, some part related to the image acquisition technique used in the field. This has been corrected for and expanded on in the Abstract, Methods, Results and Discussion sections accordingly.

For example, in the Methods we have corrected and provided further details on the Ring-MIC protocol:

“For validation purposes, a physical ring was placed on the top of the canopy for counting the number of ears in the exact ring area by visual inspection in the field. The ring has a radius of 0.1225 m. The ring was attached by an extension arm to the monopod used to acquire the RGB images. Thermal and RGB images were acquired at the same time than the visual (in-situ) ear counting (inside the ring) was assessed in the first block (i.e. 24 plots) of the rainfed trial at Valladolid. Visual counting was performed always by the same person at the same position where the images were acquired. Approximately 15 seconds were spent for each counting using a clicker to keep track of the exact number and making sure to inspect the area inside the ring in order to accurately include all ears present by moving plants and changing perspective angles at each location.”

Again, in the Discussion, the issues surrounding the Ring-MIC are more thoroughly confronted and discussed (new text highlighted):

“For additional thermal image algorithm validation purposes, visual in-situ counting was developed using a ring to delimit a specific area over the crop while in the field and thus facilitate the manual counting. Although the ring has a small area, compared with the complete plot size, we have obtained a relatively low R2 relationship against thermal image-based counting (R2 = 0.40, Fig. 4). This is most likely associated with the limited single image-perspective of the one zenithal/nadir thermal image or RGB image captured in the field. Some portion of the error could additionally be associated with human visual inspection errors in the field and potentially the subjectivity of the observer, as are often assumed to be major sources of error in manual ear counting in actual breeding programs; however, for this study the researchers have attempted to minimize the human error associated with the Ring-MIC ear counts in order to provide quality validation data. In the manual in-situ counting in the field, it was necessary to both view the canopy from different angles as well as physically move plants in order to acquire accurate field validation data, representing a major difference between the in-situ counting and the single image-perspective remote sensing approach of the automatic thermal image ear counting technique presented here. In previous studies on ear recognition no information regarding correlation between in-situ visual ear counting and automatic ear counting was provided [6–14], but it is an important point to consider as the entire image acquisition and processing pipeline represents a sum of errors. However, we have obtained good results using thermal imagery for ear counting with positive and strong relationships between the automatic thermal ear counting system and the manual image-based ear counting (R2 = 0.84 for Ring-MRC, Fig. 4; R2 = 0.75 for Ring-MTC, Fig. 4 and R2 = 0.80 for Complete-MTC, Fig. 5). Furthermore, in all comparisons, the slope of the correlation is quite close to a 1:1 ratio, indicating very little bias toward over or under-counting within the range of ear density in this study. Thus, the additional validation results provide support for the capacity of the automatic thermal image counting algorithm to count the ears that are present in the image with high precision and low bias. Still, other potential sources of error in the thermal image counting pipeline should be considered in more detail.”

Introduction: The results of the article cannot be in the introduction, so that Table 1 (line 37) must be substituted by a previous reference proving that the ear T is higher that leaf T (maybe [25]?). However, in the case of the TIR camera it seems that it is only true in clear sky conditions, so, do the results of the article contradict this assertion?

We have modified this sentence in order to clarify that this statement is indeed based on a previous study (article citation [25]), including more details regarding the referenced experimental setup for thermal acquisition, sky conditions and temperature leaf/ear conclusions. Table 1 has been removed from the introduction and the information substituted by a reference [25]. The modified version of this sentence now reads as follows:

“However, previous studies have shown that regardless of the water conditions during the growing season there are often significant constitutive differences between leaf and ear temperature on sunny days [25], with ear temperature being higher than leaf ”

2.2 Thermal images: It must be indicated the spatial resolution of the sensor (pixel size or GSD) here, not only in Table 1. Also, if there are only one channel 8-14 micrometres or several in this range. And, if there are several, the wide of each one in micrometres.

We have added the spatial resolution from de camera specifications and we have also clarified the channel used for the MIDAS infrared camera.

Line 136: I suppose that it is an error and it should be “above” instead “under”. If not, clarify.

Reviewer is right. The sentence has been corrected.

Line 137: between 0.8 and 1 m distance ¿above the canopy? Clarify. It is better explained for the RGB camera.

We have clarified the thermal image acquisition protocol used in field, “above” has been inserted in the previous comment.

Line 138: I suppose that the hours are UTC, but it should be clarified: 12:00 h UTC or similar. Respect to this, the same hours should have been chosen in the 3 stations, why they are different? Discussion: How the hour (solar angle) affects the thermal and RGB measurement?

We UTC time reference was added at the data acquisitions protocol. We have extended the discussion related to shadow effects (solar noon plus or minus 3 hours in order) as follows:

“It is still recommended that for any passive remote sensing imaging technique, the images have to be acquired within a few hours of solar noon in order to reduce shadowing and sun angle effects.“

We have also added, apart from the changes above, the necessary differences between leaf and ears temperatures for the thermal infrared spectroscopy to work well. It is for this reason that the actual time of the data acquisition at each location was slightly different-to allow for the adequate contrast between the leaves and ears in the thermal images.

Line 174: It must be indicated the spatial resolution of the sensor (pixel size or GSD.

We used the technical information for the MIDAS camera to calculate the GSD as around 0.14 cm/pixel.

Figure 3: The ring radius is different in 3a and 3b, please make zoom in 3a in order to compare better both images.

We have made a zoom in the Fig. 3a so that the two are more easily compared.

Lines 205-206: The manual RGB and thermal counting, was made independently? i.e., only observing each image or both images at the same time?

Yes, the manual thermal and RGB image-based counting were made independently. The RGB image information was used in certain occasions at the image-based counting to mark a ‘thermal ear’ if its inclusion in the ring or not was unclear, as there was a minor difference in the perspective of the two cameras.

Results 3.1: It clear that the results manual-RGB and manual-thermal versus algorithm are good (R2=0.75-0.84) and also manual-RGB vs manual-thermal (0.83) (if they were made independently). This indicates that the automatic algorithm is good, and it is a good result, the most important of this article. And it should be noted in abstract, conclusions, etc.

We have added in the Abstract, Discussion and Conclusion sections reference to the correlation results of Ring-MTC and Ring-MRC vs Ring-ATC (R2 = 0.75-0.84) and the relationship between the thermal and the RGB manual counting (R2 = 0.83). This is in addition to the relationship between Ring-ATC and Ring-MIC (R2 = 0.40) and the complete-MTC (R2 = 0.80). For example, the entire second part of the Abstract has been rewritten as follows in order to more directly note all of the relevant results:

The relationship between the thermal counting values and the in-situ visual counting was fairly weak (R2 = 0.40), which highlights the difficulties in estimating ear density from one single image-perspective. However, the results show that the automatic thermal ear counting system performs quite well in counting the ears that do appear in the thermal images, exhibiting high correlations with the manual image-based counts from both thermal and RGB images in the sub-plot validation ring (R2 = 0.75-0.84). Automatic ear counting also exhibited high correlation with the manual counting from thermal images when considering the complete image (R2 = 0.80). The results also show a high correlation between the thermal and the RGB manual counting using the validation ring (R2 = 0.83). Methodological requirements and potential limitations of the technique are discussed.”

The Conclusions, now completely re-written, read as follows:

In our study, a thermal camera was used to develop an image processing system for automatic ear counting in field conditions. In favour of the thermal counting approach, ear density values estimated through thermal imaging can be processed much more rapidly as the size of the images is much smaller compared to high resolution RGB images used in other previous studies, while the increase in contrast allows for equally accurate assessments when the thermal images are captured under specific conditions. There should be a difference of at least 2 ºC between the ear and leaf temperatures for this thermal ear counting algorithm to work. Though the correlation with manual in-situ ear counts (Ring-MIC) was not very high, the algorithm did demonstrate high correlations with various manual image-based ear counts (Ring-MRC, Ring-MTC, Complete-MTC). In future applications, thermal imagery may be acquired from multiple perspectives (including off-nadir and oblique), or even thermal video data, for improved ear detection in comparison with in-situ counts. Still, further studies could use the same thermal image segmentation algorithm developed here for ear detection (Fig. 2) in order to extract the temperature of ears and leaves separately for other phenotyping applications related to plant water stress effects or grain yield prediction. Thermal and RGB fusion, along with 3D imaging, could be the next steps for cereal ear counting in field conditions in order to take maximal advantage of the strengths of each imaging technology.

However, the result Ring-MIC vs algorithm (0.4) is low, and also the Ring-MIC vs RGB (0.37). This means that both methods (RGB and thermal cameras) are not good for counting compared to the usual MIC (assuming that this is the reference value). The authors indicate problems with the low distance in the thermal camera, but, which is the problem with the RGB camera? Is there problem also with it? Because the result is even worse. If not problem with the RGB camera, the results had been similar with the thermal camera, I suppose.

In part as a response to Reviewer 1 as well as this specific comment, we have also added new reasoning regarding the human vision inspection of the Ring-MIC and the single thermal and RGB image perspective and have incorporated it into the Discussion and Conclusions accordingly. We consider the results of 0.40 and the 0.37 to be both fairly weak but on the same scale. The additional reasoning presented is not something specific to either thermal or RGB, but rather the use of one single perspective (an image, whether RGB or thermal) for counting ears that are in essence distributed in 3D space. We have made sure that the details in the protocol provided as well as the Discussion and Conclusions represent this additional reasoning on the source of errors and potential for improvements in future work.

In the methods section:

For validation purposes, a physical ring was placed on the top of the canopy for counting the number of ears in the exact ring area by visual inspection in the field. The ring has a radius of 0.1225 m. The ring was attached by an extension arm to the monopod used to acquire the RGB images. Thermal and RGB images were acquired at the same time than visual (in-situ) ear counting (inside the ring) were assessed in the first block (i.e. 24 plots) of the rainfed trial at Valladolid. Visual counting was performed always by the same person at the same position where the images were acquired. Approximately 15 seconds were spent for each counting using a clicker to keep track of the exact number and making sure to inspect the area inside the ring in order to accurately include all ears present by moving plants and changing perspective angles at each location.”

In the Discussion:

This is most likely associated with the limited single image-perspective of the one zenithal/nadir thermal image or RGB image captured in the field. Some portion of the error could additionally be associated with human visual inspection errors in the field and potentially the subjectivity of the observer, as are often assumed to be major sources of error in manual ear counting in actual breeding programs; however, for this study the researchers have attempted to minimize the human error associated with the Ring-MIC ear counts in order to provide quality validation data. In the manual in-situ counting in the field, it was necessary to both view the canopy from different angles as well as physically move plants in order to acquire accurate field validation data, representing a major difference between the single image-perspective remote sensing approach of the automatic thermal image ear counting technique presented here. In previous studies on ear recognition, no information regarding the correlation between in-situ visual ear counting and automatic ear counting was provided [6–14], but it is nonetheless an important point to consider as the entire image acquisition and processing pipeline represents a sum of errors. Of course, the approach for visual counting assayed was in fact much faster than the traditional ear counting procedures, which implies for example counting the total number of ears in one-meter linear row length. However, this approach is quite tedious (and of course takes much longer than the 15 s per plot as in our study).

In the Conclusions:

Though the correlation with manual in-situ ear counts (Ring-MIC) was not very high, the algorithm did demonstrate high correlations with various manual image-based ear counts (Ring-MRC, Ring-MTC, Complete-MTC). In future applications, thermal imagery may be acquired from multiple perspectives (including off-nadir and oblique), or even thermal video data, for improved ear detection in comparison with in-situ counts.

On the other hand, in the validation, one must also indicate always not only R2, but also RMSE and bias as minimum for each case.

We have indicated the RMSE and bias for the regression in the Figures 4 and 5.

Minor problems to correct/explicate to me:

Lines 100-102: Reference for this data (meteorological data at the stations, AEMET data, etc.?

We have added the information system (SIAR, http://eportal.mapama.gob.es/websiar/Inicio.aspx) used to obtain the meteorological data, we have also included the webpage in the references.

Line 113: I do not understand “replicates”. Are 24 plot per station and they were measurement 3 times each one and the obtained the mean value?

The replicates are that there were 24 genotypes planted in three rows, for a total of 3x24=72 plots per trial location. We have added for clarity “(3 replicates x 24 genotypes)” at this location before indicating that the total was 72 total plots.

Lines 117-119: Same comment that lines 100-102. In the case of Sevilla, it is 2016/2017 instead 2015/2017, is not?

The first thermal image acquisition was carried out at the crop season 2016/2017 in Sevilla. The text the manuscript is correct and no changes have been made.

Lines 150-151 and lines 165-167: Which is the color threshold used? Or the H, S and V thresholds used? And in terms of T in kelvin?

We have added information regarding color ranges (from green to red colour) in order to clarify the Hue values used from the HSV colour space.

“…represented in colours between red and green, that correspond to Hue values from 2º to 120º,…

Figures 4 and 5: Increase the text size.

The text size was modified. We have also added the 1:1 line in each figure.

Figure 4 and Figure 5 footnotes: Always it is “y” vs “x”, not “x” vs “y”.

The order has been corrected at the figure legends (Fig 4 and 5).

We appreciate the detailed comments that the reviewer has provided towards the improvement of this manuscript. While it is indeed true that not all of the results of this study were strong, we feel that there is strong support for the functionality of the presented algorithm for analyzing high resolution thermal images, the focus of this special issue. Moreover, we have included an expanded analysis on the potential sources of error in the entire imaging and analysis pipeline and have included relevant suggestions for improvement as future work. It our hope that the Reviewer finds that these changes with regards to each specific point have resulted in an acceptable manuscript for publication in this special issue of Remote Sensing.

Round 2

Reviewer 1 Report

This is an improved version of a previously submitted manuscript. The authors have reworked the manuscript and amended it according to the reviewers suggestions. A minor point remains at Line 275-278. Consider rephrasing, sentence is unclear. The original comments still stands because the edited version of this sentence is still (really) unclear. Could this be simplified? If not, I would consider removing it altogether.

Author Response

Response to Reviewer 1:

Comments to the Author

This is an improved version of a previously submitted manuscript. The authors have reworked the manuscript and amended it according to the reviewers suggestions. A minor point remains at Line 275-278. Consider rephrasing, sentence is unclear. The original comments still stands because the edited version of this sentence is still (really) unclear. Could this be simplified? If not, I would consider removing it altogether.

We have again simplified the phrasing of this sentence to make it more clear:

“The increase in contrast provided by thermal imaging stems from large differences in ear and leaf transpiration rates, which directly affect cooling capacity and temperature.”

Reviewer 2 Report

See attached file.

Author Response

Response to Reviewer 2:

Comments to the Author

The article has improved significantly and now I only have minor comments to correct/clarify. I write below my previous comments and your response and again my comments now:

Line 138: I suppose that the hours are UTC, but it should be clarified: 12:00 h UTC or similar. Respect to this, the same hours should have been chosen in the 3 stations, why they are different?

We UTC time reference was added at the data acquisitions protocol.

I do not understand this. When you put 12:00 UTC+2, what do you mean? (lines 149-150 in the new paper) The local solar noon is 12:00 UTC and, in summer, this is 12+2=14 h in the official hour in Spain (hour of a Spanish watch). So, if this measurement was done at local solar noon you have to put 12:00 UTC simply (and it will be at the 14 h in the watch). If you put 12:00 UTC+2, I understand that the measurement was done at the 14 h+2 h=16 h of a Spanish watch. Which is the case? You must use only the notation UTC, without +nothing.

Response: We have corrected the UTC notation using this explanation kindly provided by the reviewer.

We have extended the discussion related to shadow effects (solar noon plus or minus 3 hours in order) as follows:

I agree. But then, in Valladolid, with 16:30-17:00 UTC, there are more than 3 hours, there is 4,5-5 h after local solar noon and your criteria is not complied. It is necessary an explanation why these measurements were taken so later.

Response: We have also added the UTC notation at the ‘3.2. Understanding acquisition and algorithm errors’ section in order to clarify the acquisition time. With the correction of the UTC notation errors fixed above now it is clear that all of our measurements were indeed taken with the +/- 3 hours guidelines.

We have also added, apart from the changes above, the necessary differences between leaf and ears temperatures for the thermal infrared spectroscopy to work well. It is for this reason that the actual time of the data acquisition at each location was slightly different-to allow for the adequate contrast between the leaves and ears in the thermal images.

This could be your explanation...I do not know. If it is true, it should be indicated in the text, that the difference between time of the data acquisition in Valladolid is due to this the necessity of finding differences between leaf and ears temperatures. This sentence is good “It is for this reason that the actual time of the data acquisition at each location was slightly different-to allow for the adequate contrast between the leaves and ears in the thermal images”, but now it is not in the article.

Response: We have added this sentence at the ‘2.2. Thermal images’ section.

---------------------------------------------------------------------------------------------------------------

Line 174: It must be indicated the spatial resolution of the sensor (pixel size or GSD.

We used the technical information for the MIDAS camera to calculate the GSD as around 0.14 cm/pixel.

I refer in this line to the RGB camera, Sony camera.

Response: We have also added the spatial resolution for the RGB Sony camera at the ‘2.4.1. Manual in-situ counting and RGB images’ section.

---------------------------------------------------------------------------------------------------------------

However, the result Ring-MIC vs algorithm (0.4) is low, and also the Ring-MIC vs RGB (0.37). This means that both methods (RGB and thermal cameras) are not good for counting compared to the usual MIC (assuming that this is the reference value). The authors indicate problems with the low distance in the thermal camera, but, which is the problem with the RGB camera? Is there problem also with it? Because the result is even worse. If not problem with the RGB camera, the results had been similar with the thermal camera, I suppose.

In part as a response to Reviewer 1 as well as this specific comment, we have also added new reasoning regarding the human vision inspection of the Ring-MIC and the single thermal and RGB image perspective and have incorporated it into the Discussion and Conclusions accordingly. We consider the results of 0.40 and the 0.37 to be both fairly weak but on the same scale. The additional reasoning presented is not something specific to either thermal or RGB, but rather the use of one single perspective (an image, whether RGB or thermal) for counting ears that are in essence distributed in 3D space. We have made sure that the details in the protocol provided as well as the Discussion and Conclusions represent this additional reasoning on the source of errors and potential for improvements in future work.

OK with all the added text. But I also asked if the distance above the canopy for the RGB camera is the same that the distance for the thermal camera in Valladolid, where you had problem with the error in the distance, Fig. 6, where you say that the thermal image was acquired at less than 0.8 m distance. Is the same for the RGB camera, <0.8 m?

Response: We have also added the spatial resolution for the RGB Sony camera at the ‘2.4.1. Manual in-situ counting and RGB images’ section.

Regarding problems with errors in the distance (Fig. 6c), the main problem is related with the thermal camera technology itself and the acquisition distance protocol. While the focal length of the MIDAS thermal camera is adjusted manually; the Sony RGB camera allows the autofocus option, avoiding out-of-focus images by distance changes in field conditions. On the other hand, and regardless the technology, thermal or RGB images database must be acquired using the same distance, although RGB cameras are less prone to acquire out-of-focus images than thermal cameras, the image processing system requires uniform images from the acquisition step and changes in the protocol (Fig. 6c) will cause errors in the automatic system. In addition, the single perspective issues are the same for thermal and RGB images for ear detection.

--------------------------------------------------------------------------------------------------------------

On the other hand, in the validation, one must also indicate always not only R2, but also RMSE and bias as minimum for each case.

We have indicated the RMSE and bias for the regression in the Figures 4 and 5.

These values must be included inside the Figures, not in the footnote. I mean, where you had put R2 and p- value, inside each figure. And please, also increase the text size insides the figures.

Response: We have included the RMSE and bias and increase text size inside each figure.

-------------------------------------------------------------------------------------

Minor problems to correct/explicate to me:

Lines 117-119: Same comment that lines 100-102. In the case of Sevilla, it is 2016/2017 instead 2015/2017, is not?

The first thermal image acquisition was carried out at the crop season 2016/2017 in Sevilla. The text the manuscript is correct and no changes have been made.

Then we agree. In the text is wrote 2015/2017 and it should be 2016/2017, as I told you and you confirm.

Response: Thank you kindly for this correction.

-------------------------------------------------------------------------------------------------------------------

Lines 150-151 and lines 165-167: Which is the color threshold used? Or the H, S and V thresholds used? And in terms of T in kelvin?

We have added information regarding color ranges (from green to red colour) in order to clarify the Hue values used from the HSV colour space.

“...represented in colours between red and green, that correspond to Hue values from 2º to 120º,...

The correct unit is 2º or 200º, put well the units. Values of 120º are extremely high, maybe it is an error? As high temperature values are impossible. If not, clarify.

Response: We have deleted the “ º ” units in order to clarify the Hue values used from the HSV color space derived from the thermal colour map in the automatic system.

-------------------------------------------------------------------------------------------------------------------
